

# Tangent-space methods for truncating uniform MPS

**Bram Vanhecke[*], Maarten Van Damme, Jutho Haegeman,
Laurens Vanderstraeten and Frank Verstraete**

Department of Physics and Astronomy, University of Ghent,
Krijgslaan 281, 9000 Gent, Belgium

[*] bavhecke.Vanhecke@UGent.be

## Abstract

An essential primitive in quantum tensor network simulations is the problem of approximating a matrix product state with one of a smaller bond dimension. This problem forms the central bottleneck in algorithms for time evolution and for contracting projected entangled pair states. We formulate a tangent-space based variational algorithm to achieve this goal for uniform (infinite) matrix product states. The algorithm exhibits a favourable scaling of the computational cost, and we demonstrate its usefulness by several examples involving the multiplication of a matrix product state with a matrix product operator.

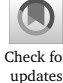

The density matrix renormalization group (DMRG) [1,2] and quantum tensor networks [3, 4] provide algorithms for simulating ground states of strongly correlated quantum many body systems with a computational cost that scales linear in the system size, thereby overcoming the infamous exponential wall of the quantum many body problem. The physical parameter controlling the computational cost is the entanglement entropy, as directly reflected in the bond dimension $\chi$ of the corresponding matrix product states (MPS) [5]. However, there are many interesting physical problems for which this bond dimension can become prohibitively large, such as the problem of simulating time evolution of a quantum state out of equilibrium or of contracting a tensor network comprised of a projected entangled pair state (PEPS) with a large bond dimension. In both cases, the central problem is to approximate the product of an MPS and a matrix product operator (MPO) with an MPS of smaller bond dimension. For both finite and infinite systems, a well-known algorithm to achieve this is time-evolving-block-decimation and variations thereof [6–9], all of which rely on a local truncation of Schmidt values, therefore not being optimal for the global wavefunction. For finite systems, a considerable improvement over those algorithms can be obtained by adopting a variational approach which optimizes the fidelity by sweeping through the system and solving alternating linear problems [10,11]. The computational cost of the latter algorithm has a better scaling as it does not require bringing the joint MPS/MPO system in canonical form, and furthermore achieves a better overal fidelity due to its variational nature.

In this paper, we present the uniform and infinite version of that algorithm. It is based on ideas developed in the context of tangent space methods for uniform matrix product states [12,

13] and the variational uniform matrix product state (vumps) algorithm [14, 15]. Our main motivation is the development of efficient MPS algorithms which can deal with time-evolution methods involving MPOs with large bond dimension and of efficient and well-conditioned ways of contracting PEPS [16]. It also overcomes a main limitation of algorithms based on the time-dependent variational principle (TDVP) [17–19], where it is difficult to build up entanglement starting from a low-entangled state by allowing large time steps.

The paper is organized as follows. In the first section we discuss how to approximate a given uniform MPS variationally with another one with smaller bond dimension. In a second section, we illustrate this algorithm with several relevant examples.

*Fixed-point equations.*—We start from the diagrammatic expression of a uniform MPS in the thermodynamic limit, parametrized by a single tensor $A$

$$|\Psi(A)\rangle = \text{—} \boxed{A} \text{—} \boxed{A} \text{—} \boxed{A} \text{—} \boxed{A} \text{—} \boxed{A} \text{—} \, . \tag{1}$$

We will assume a trivial unit cell in this text for simplicity, the case of larger unit cells is treated straightforwardly. Using the gauge freedom of the MPS we can choose this tensor to be in the left canonical gauge $A_L$ or the right canonical gauge $A_R$, with

$$\tag{2}$$

These gauge-fixed tensors are related by a matrix $C$ as

$$\boxed{A_L} \text{—} \boxed{C} = \boxed{C} \text{—} \boxed{A_R} = \boxed{A_C} \, , \tag{3}$$

allowing us to bring the MPS into the so-called mixed gauge

$$|\Psi(A)\rangle = \text{—} \boxed{A_L} \text{—} \boxed{A_L} \text{—} \boxed{A_C} \text{—} \boxed{A_R} \text{—} \boxed{A_R} \text{—} \, . \tag{4}$$

For a given MPS $|\Psi(M)\rangle$ described by a tensor $M$, we now wish to find an MPS $|\Psi(A)\rangle$ such that the latter approximates the former in some optimal way. A natural choice for an optimality condition is that they should have a maximal fidelity, which leads us to a variational optimization problem for the tensor $A$:

$$\arg\max_{A,A^*} \left( \log\left[ \frac{\langle\Psi(A^*)|\Psi(M)\rangle \, \langle\Psi(M)|\Psi(A)\rangle}{\langle\Psi(A^*)|\Psi(A)\rangle} \right] \right) . \tag{5}$$

To have a properly defined cost function, we consider the logarithm of the fidelity, which is an extensive quantity that scales with the system size in the thermodynamic limit, and replace the cost function by its intensive version, *i.e.* the density of logarithmic fidelity, instead. Equivalently, this amounts to replacing the norm in the expression for the fidelity with a modified norm

$$\lambda = \lim_{N\to\infty} \left( \langle\Psi(A^*)|\Psi(M)\rangle_N \right)^{1/N} , \tag{6}$$

where $\langle\Psi(A^*)|\Psi(M)\rangle_N$ represents the overlap of two MPS of length $N$ with periodic boundary conditions, made up of tensors $M$ and $A$. This limit is unique and well defined if both $M$ and $A$ are injective MPS tensors, and converges to the largest eigenvalue $\lambda$ of the mixed transfer matrix,

$$\lambda = \lambda_{\max} \left( \begin{array}{c} \text{—} \boxed{M} \text{—} \\ | \\ \text{—} \boxed{A^*} \text{—} \end{array} \right) . \tag{7}$$

This cost function being a real-valued function of the tensor $A$ and its conjugate $A^*$, the gradient is obtained by differentiating the cost function with respect to $A^*$. An optimal point is reached when the gradient vanishes,

$$\langle \partial_{A^*} \Psi(A^*) | \left( | \Psi(M) \rangle - \frac{\langle \Psi(A^*) | \Psi(M) \rangle}{\langle \Psi(A^*) | \Psi(A) \rangle} | \Psi(A) \rangle \right) = 0. \tag{8}$$

Here, $|\partial_A \Psi(A)\rangle$ can be interpreted as a tangent vector on the manifold of MPS [12,13]. The MPS tangent space contains the state $|\Psi(A)\rangle$ itself, which represents the direction of infinitesimal changes in phase or normalization of the tensor. However, inserting this direction in Eq. 8 yields a trivial equation, exactly because our cost function is insensitive to such changes in phase or normalization. Using the projector $\mathcal{P}_A$ onto the part of tangent space that is orthogonal to $|\Psi(A)\rangle$, the optimality condition can be reformulated as

$$\mathcal{P}_A |\Psi(M)\rangle = 0. \tag{9}$$

Note that the right hand side of Eq. 9 thus vanishes, but it will prove propitious to keep track of it. An explicit form of $\mathcal{P}_A$ in the mixed-gauge is given by [13]

$$\mathcal{P}_A = \sum_{i \in \mathbb{Z}} \quad \tag{10}$$

Applying this operator to $|\Psi(M)\rangle$, which we assume to be a uniform MPS parameterized by a single tensor $M$, we find that the optimality condition [Eq. (9)] is satisfied if and only if

$$A'_C = A_L C' = C' A_R, \tag{11}$$

where $A'_C$ and $C'$ are given by

$$\quad = \lambda \; A'_c \; , \tag{12}$$

and

$$\quad = \; C' \; , \tag{13}$$

with the fixed points $G_L$ and $G_R$ given by the eigenvalue equations

$$\quad = \lambda \; G_L \; , \qquad \quad = \lambda \; G_R \; , \tag{14}$$

where as before $\lambda$ is the largest eigenvalue of the mixed transfer matrix that appears here (both with the choice $A = A_L$ and $A = A_R$, but this does not affect the eigenvalue as they are related by a similarity transform). Eq. (11) can only be satisfied if $C' \sim C$ and $A'_C \sim A_C$.

This observation motivates an algorithm where, starting from a randomly initialised tensor $A$, we identify the resulting $A'_C$ and $C'$ as the new target values of $A_C$ and $C$. As one cannot

---

**Algorithm 1** Variationally optimizing overlap of uniform MPS with trial state $|\Psi(M)\rangle$

1: bring $A$ in canonical form $\{A_L, A_R\}$
2: **repeat**
3:     compute $\lambda$, $G_L$ and $G_R$                                                  ▷ Eq. (14)
4:     find new $A_C'$ and $C'$                                                           ▷ Eqs. (12)-(13)
5:     extract new $A_L$ and $A_R$                                                        ▷ Eq. (15)-(16)
6:     compute error $\epsilon$                                                           ▷ Eq. (17)
7: **until** $\epsilon < \eta$
8: **return** $A_L, A_R, \lambda$

---

define an MPS via the center-site tensors directly, one crucial step in each iteration will be the extraction of a new set of MPS tensors $\{A_L, A_R\}$ from the $A_C'$ and $C'$ that were obtained. A close-to-optimal solution of this problem is given by the prescription [13]

$$A_L \leftarrow U_l V_l^{\dagger}, \quad \begin{cases} A_C' = U_l P_l \\ C' = V_l Q_l \end{cases} \tag{15}$$

and

$$A_R \leftarrow U_r^{\dagger} V_r, \quad \begin{cases} A_C' = P_r U_r \\ C' = Q_r V_r \end{cases}, \tag{16}$$

where all decompositions involve unique polar decompositions or their transposed. This approach is similar to the one adopted in the standard vumps algorithm [15]. Once we have obtained a new set $\{A_L, A_R\}$, we can re-compute the fixed-point tensors $G_L$ and $G_R$ and the scheme can be reiterated. As a convergence measure we take the norm of the fixed-point equation in Eq. 11, which is given by

$$\epsilon = \left| -\!\!\boxed{A_C'}\!\!- \quad - \quad -\!\!\boxed{A_L}\!\!-\!\!\boxed{C'}\!\!- \right|, \tag{17}$$

where $A_C'$ and $C'$ are given by Eqs. 12 and 13; this convergence measure becomes zero only when the gradient [Eq. 8] vanishes and therefore characterizes a variationally optimal approximation.

A specific instance of the above scheme occurs when applying a uniform matrix product operator (MPO) to a given MPS, and approximating the resulting state as an MPS with a certain bond dimension. In that case the above fixed-point equations are given by

$$\tag{18}$$

and

$$\tag{19}$$

with

$$
G_L \begin{array}{c} M \\ O \\ A_L^* \end{array} = \lambda \; G_L \;, \qquad \begin{array}{c} M \\ O \\ A_R^* \end{array} G_R = \lambda \; G_R \;.
\tag{20}
$$

Our variational method can, therefore, be used for approximating an MPS-MPO state by an MPS with the original bond dimension of $M$. This is an operation that appears in many MPS methods (see further), and we can show that our approach scales more favourably as compared to the standard local-truncation approach [9]. Indeed, supposing that both the original and new MPS have bond dimension $\chi$ and physical dimension $d$ and the MPO has bond dimension $D$, the time-complexity of the above scheme is $\mathcal{O}(\chi^3 D d + \chi^2 D^2 d^2)$, and the memory required scales as $\mathcal{O}(\chi^2 D d)$. We can compare this to the complexity of cutting the bond dimension by truncating local Schmidt values. The most costly operation required to cut the bond this way is following contraction:

$$
\rho \begin{array}{c} M \\ O \\ O^* \\ M^* \end{array} = \tilde{\rho} \;.
\tag{21}
$$

The time-complexity of this operation is $\mathcal{O}(\chi^3 D^2 d + \chi^2 D^3 d^2)$ and the memory required $\mathcal{O}(\chi^2 d D^2)$. In addition, one typically performs a full singular-value decomposition of a square $\chi D$ matrix, for which the time complexity scales as $\mathcal{O}(\chi^3 D^3)$. This analysis shows that for MPOs of large virtual dimension $D$, the method we prescribe can be a significant, even crucial, improvement.

*Truncating an MPS.*—Let us first illustrate this variational method by truncating the bond dimension of a given MPS. Again, the most commonly used technique for that purpose is the truncation of local Schmidt values on all bonds simultaneously [7] which is not optimal for the global wavefunction. We compare the two techniques in Fig. 1 for an MPS of considerable dimension. We find that local truncation performs fairly well across the board, but that our variational scheme still finds a slightly better state after convergence. This example shows that our fidelity optimization can be useful only if precision is of the utmost importance.

*Time evolution.*—There are roughly two different classes of methods used to time-evolve an infinite MPS. The first class tries to directly transform the Schrödinger equation into a (non-linear) differential equation on the variational manifold. This is exactly the mechanism behind TDVP [17,18], where the direction in which the state needs to change (the right hand side of the Schrödinger equation) is projected onto the tangent space of the MPS. The second class of methods instead starts from an approximation of the time evolution operator $\exp(-iH\delta)$ for a certain time step $\delta$. This approximation is provided in terms of a quantum circuit, or, more generally, an MPO, and can be obtained from e.g. a Suzuki-Trotter decomposition [6,21,22] or other schemes [16,23]. The resulting MPO is then applied to the current state, encoded as MPS, followed by a bond truncation[1]. With a (low-order) Suzuki-Trotter decomposition, the MPO bond dimension can remain low, but feasible time steps $\delta$ are also very small. With the cluster expansion from Ref. [16], it is easier to reach larger $\delta$, at the cost of a higher MPO bond dimension. It is therefore infeasible to apply this MPO to an MPS and truncate directly

---

[1]Note that methods based on Krylov subspaces or Taylor expansions of the evolution operator, which are common for time-evolving finite MPS, do not work in the thermodynamic limit because they are not extensive.

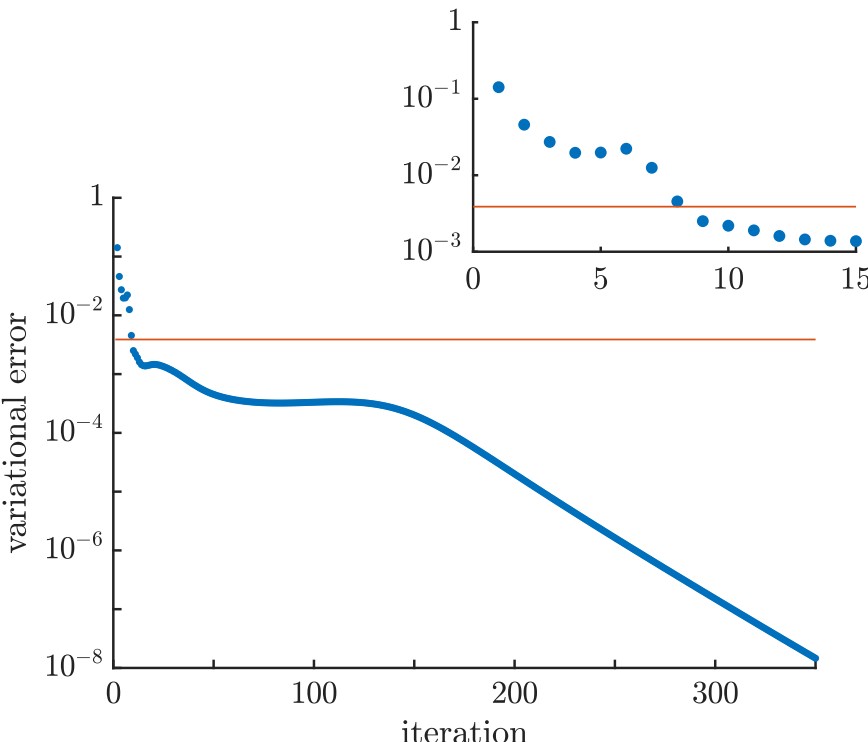

Figure 1: Truncating an MPS to a lower bond dimension. We show the variational error $\epsilon$ [Eq. (17)] in each iteration of the fidelity optimization (blue), compared to that same variational error measure computed for the state obtained by a local singular-value truncation (red). After eight iterations the variational error is smaller, but we can converge a lot further using our iterative scheme. The fidelity per site $\lambda$ with the original state is $1 - 5.37 \times 10^{-5}$ and $1 - 3.78 \times 10^{-5}$ respectively, showing that we can improve the state with our variational scheme. The starting MPS is an SU(2)-symmetric ground-state approximation for the spin-4 Heisenberg model (which has very large correlation length [20]) with 13 charge sectors and maximal bond dimension in each sector $D_{\text{max}} = 512$, yielding a total bond dimension of $D_{\text{total}} \approx 21600$; this state was obtained using the vumps algorithm. The truncated MPS has 8 charge sectors with $D_{\text{max}} = 27$, yielding a total bond dimension of $D_{\text{total}} \approx 700$.

according to the Schmidt values due to prohibitive memory constraints or time complexity considerations. In this case thus, our method is indispensable. This leads to a time evolution scheme that we here simply state as a possible application of our truncation method, but is discussed in more detail in [16].

We illustrate this usage by evolving the Néel state with the XXZ Hamiltonian.

$$H_{\text{XXZ}} = \sum_i S_i^x S_{i+1}^x + S_i^y S_{i+1}^y + \Delta S_i^z S_{i+1}^z,$$

where $S_i^\alpha$ the spin-1/2 operators at site $i$ and we choose $\Delta = 1/2$. This problem is closely related to the one considered in Ref. [24] asserting the supremacy of quantum simulators. We have exploited the $U(1)$ symmetry of the system and used an MPS with a two-site unit cell and a maximal bond dimension of 994. The MPO bond dimension is 21, which enabled an accurate time step of up to $dt = 1.2$. In Fig. 2 we show the offset of the staggered magnetization from its initial maximal value —as measured by $(\mathbb{1} + Z)/2$ on one of the two sites in the unit cell– as a function of time, and benchmark it with a simulation with the TDVP algorithm, where we manually expand the bond dimension with small noise [25].

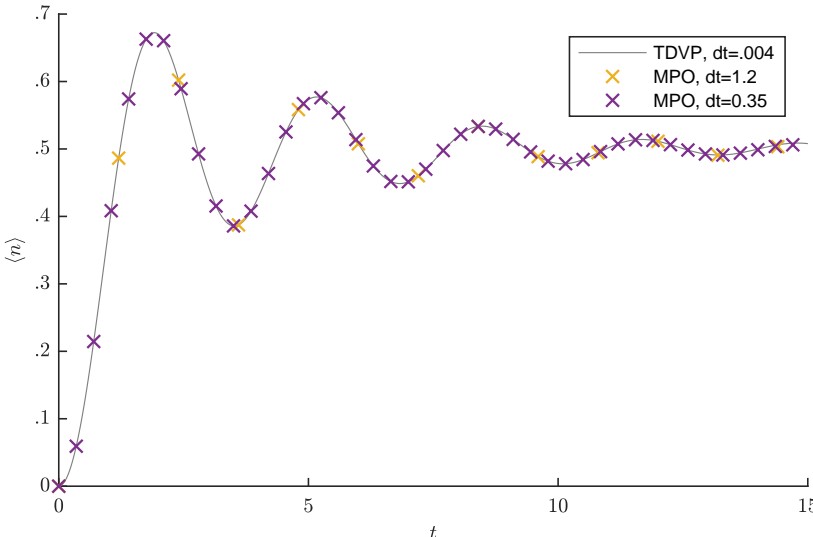

Figure 2: Time evolution of the staggered magnetization relative to its initial maximal value, as measured by $(\mathbb{1}+Z)/2$ on one of the sites, for the Néel state evolved with the XXZ Hamiltonian with $\Delta = 1/2$. We show results for different time steps for the MPO cluster expansion from Ref. [16]. The gray line is a reference result obtained with TDVP with very small time step. We have made explicit use of the $U(1)$ symmetry, and fixed the total bond dimension to $\chi = 994$.

*Power method for transfer matrices.*—Let us now consider the calculation of an MPS fixed point of an MPO transfer matrix by way of the power method: In each iteration we apply the MPO and truncate the bond dimension, until the MPS converges to a fixed point. Power methods have been used for computing transfer fixed points where the local singular-value truncation was adopted in each iteration [9], but here we use our variational truncation. In contrast to the former, the fixed point of our variational-truncation approach is, in fact, a variationally optimal MPS in the sense that it optimizes the leading eigenvalue for hermitian transfer matrices. Indeed, in the fixed point of this power method, the top-layer MPS in the fixed-point equations [Eqs. (18)-(20)] should be the same as the down-layer, and the equations reduce to the usual fixed-point equations of the vumps algorithm (which is variationally optimal for hermitian transfer matrices). Hence, both approaches share at least the same fixed point, which is not true with a scheme based on local truncations.

For hermitian transfer matrices the performance of a power method is inferior to that of the Krylov-inspired vumps algorithm [26], but it is very useful in cases of spatial symmetry breaking where the fixed point alternates between different MPSs or for non-Hermitian MPOs. We illustrate this case by studying the MPO transfer matrix of the classical antiferromagnetic Ising model on the square lattice (Fig. 3). In the (low-temperature) symmetry-broken phase, we find that the power method alternates between two MPSs that are the same up to a one-site translation. We look at different convergence criteria and also compare to the ferromagnetic fixed point (found using vumps), on which we performed a sublattice rotation (i.e., flipping the spin on every other site). The results are presented in Fig. 3.

*Dynamical growing of bond dimension.*—Our variational-truncation approach is particularly useful as a way of enlarging the bond dimension of an MPS when simulating time evolution or computing fixed points of transfer matrices. With respect to the former, the most persistent critique to the TDVP algorithm revolves around the fact that it projects the time evolution on the manifold of MPS with a fixed bond dimension, and that it is impossible to grow the

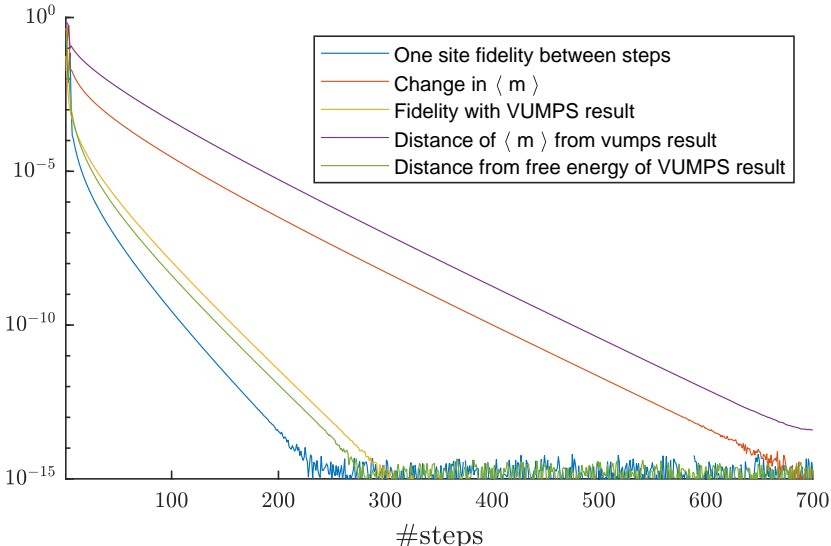

Figure 3: Different error measures to determine the convergence of the power method approach to find the MPS fixed point of the MPO transfer matrix of the antiferromagnetic Ising model at inverse temperature $\beta = 1.01\beta_c$. From top to bottom in the legend, we show (1) one minus the one site fidelity between site 1 and site 2 an iteration later, (2) the change in the local magnetization after an iteration, (3) one minus the fidelity with the sublattice rotated ferromagnetic vumps result, (4) difference of the local magnetization with the one from the vumps result, (5) difference of the free energy with the one from the vumps result.

bond dimension during the evolution. Our variational algorithm is not confined to a manifold of fixed bond dimension, because we can choose the bond dimension at each time step. We believe that a 'hybrid' between TDVP and our current scheme can provide a good way of simulating time evolution variationally using MPS where the amount of entanglement increases through time.

For fixed points of transfer matrices we can exploit our fidelity optimization in a similar way. We imagine the situation in which we have found a fixed-point MPS of a certain bond dimension, and we wish to find a better MPS of larger bond dimension. We can now use the previous MPS to construct an initial guess, apply the transfer matrix to this MPS, and then truncate to an MPS of the desired bond dimension using the equations above [Eqs. (18)-(20)]. The resulting MPS is already a more accurate approximation of the desired state than the previous one, and thus makes an excellent initial guess for running a new fixed-point algorithm at this higher bond dimension. This is especially useful in the context of PEPS algorithms, where the fixed point calculation of the PEPS double layer is the main bottleneck.

*Conclusions.*—We have discussed a method for approximating a uniform and infinite MPS by an MPS of smaller bond dimension in a way that is variationally optimal. We show that it performs slightly better in terms of accuracy as compared to the standard method in the MPS literature. Our method is proven most useful if the MPS being approximated has some substructure (e.g., being made up of an MPO times and MPS), because it has a significantly lower computational cost in that case. In this case the method has lower complexity and requires less memory than standard alternatives. We illustrate this with time evolution using an MPO that approximates the evolution operator, a power method for finding transfer matrix fixed points, and dynamical growing of bond dimension.

The generalization of this method to the (2+1)-dimensional case can easily be envisioned,

and would be interesting to investigate. An algorithm that variationally determines a PEPS approximation of some other PEPS—perhaps a projected entangled-pair operator (PEPO) times a PEPS—can readily be devised based on the algorithm in Ref. [27]. The uses of such a method would be identical to the ones presented here: performing accurate and reliable time evolution, a power method for determining fixed points of non-hermitian PEPOs or PEPOs exhibiting spatial symmetry breaking, and growing of a PEPS bond dimension.

*Acknowledgements.*—This project has received funding from the European Research Council (ERC) under the European Union's Horizon 2020 research and innovation programme (grant agreement No 647905 – QUTE and No 715861 – ERQUAF) and from the Research Foundation Flanders (grant No G087918N).

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
