# Peer review of "Tangent-space methods for truncating uniform MPS"

_SciPost Physics Core, doi:SciPost Phys. Core 4, 004 (2021)_

## Round 2 · Referee Report · Johannes Hauschild (Referee 2) · 2021-1-21

Report

The authors have properly addressed the requested changes from the previous review, so I think this paper ready for publication.

Requested changes

None

---

## Round 2 · Referee Report · Anonymous (Referee 1) · 2021-1-22

Report

The authors have addressed enough of the points brought up in my previous report: I thus recommend the publication of the manuscript in SciPost. I invite the authors to nevertheless take a look at the following comments:

  1. I appreciate the reformulation of the definition of the cost function. I still find the wording around it a bit confusing though. Among others, the sentence just before Eq. 6 seems to refer to $\langle \Psi(A^*)|\Psi(M)\rangle$ as a "norm", which it is not in the mathematical sense of the term, and this can be confusing. It would be more accurate to speak of "overlaps" or "inner products".

  2. I find the sentence right after Eq. 9 rather confusing. In the previous manuscript, the rhs of Eq. 7 was the part proportional to $|\Psi(A)\rangle$, which vanished under the projector $\mathcal P_A$, and it prompted requested change #2 in my previous report. Now, that equation is no longer there. What I understand as a reader is that Eq. 9 comes from taking the vector whose overlap with the tangent space we need to vanish, i.e. $|\Psi(M)\rangle-\dfrac{\langle\Psi(A^)|\Psi(M)\rangle}{\langle\Psi(A^)|\Psi(A)\rangle}|\Psi(A)\rangle$ from Eq. 8, applying the projector (which annihilates the part proportional to $|\Psi(A)\rangle$) and equating that to 0. The rhs of Eq. 9 arises from this last equating to zero part, i.e., it was never anything other than zero. In what sense can we "keep track of it"?

  3. I still believe the "if and only if" right before Eq. 11 (e.g. point 3 of my previous report) is not entirely obvious. In their reply, the authors mention an added comment right after Eq.17, which says that said equation vanishes only if the gradient does. Again, it is self-evident that (17) vanishing implies (9) is satisfied, while the converse, though it seems intuitively correct, could use more justification at the mathematical level (since the reader is likely unfamiliar with the mathematical standing of infinite sums and subtractions of infinite MPS). It is however a valid choice by the authors to omit said justificacion if it is technical enough that it would derail the proper flow of the paper.

  4. What does $\sim$ mean in the sentence after Eq. 14: "Eq. (11) can only be satisfied if $C'\sim C$ and $A'_C\sim A_C$"?

  5. I guess the reader might still be a bit lost regarding how significant the fidelity improvement between the two methods in Fig. 1 is in a practical sense, though I understand it may be difficult to assess it generally.

  6. In the power method section, I understand it would be cumbersome to prove that the antiferromagnetic fixed points can be found as the two sublattice rotated ferromagnetic fixed points, thus allowing the latter to be used as a benchmark for the antiferromagnet obtained via power method (which, it is to be guessed, could not be obtained via vumps as there is no single fixed point). Just to reiterate my opinion, stating this rather than implying it could ease the reader's task.

  7. Regarding #14 in my previous report, the authors's understanding probably exceeds mine in this, but I keep wondering if the difference between the improvements in computational cost for the cases with and without substructure justifies singling out the former. In the paper it is argued in detail that in the case with substructure, the cost of their scheme (time,memory) is

    $$O(\chi^3 D d + \chi^2 D^2 d), O(\chi^2Dd)$$
    versus
    $$O(\chi^3 D^2 d + \chi^2 D^3 d + \chi^3 D^3), O(\chi^2D^2d)$$
    for the local truncation scheme. If I have not made mistakes, in the no-substructure case, where we have to approximate an MPS of bond dimension $\chi_2$ by one of dimension $\chi$, their scheme would give
    $$O(\chi^2 \chi_2 d + \chi \chi_2^2 d), O(\chi_2^2d)$$
    versus
    $$O(\chi_2^3 d), O(\chi_2^2d)$$
    for the local truncation. Indeed, the savings in memory cost disappear, but in computational time they could be comparable to the other case, since $\chi_2$ could be though of being $\sim\chi D$. Once more, though, it is up to the authors to emphasize what they find more relevant in their work.

  • validity: -
  • significance: -
  • originality: -
  • clarity: -
  • formatting: -
  • grammar: -

Author:  Bram Vanhecke  on 2021-02-11  [id 1227]

(in reply to Report 2 on 2021-01-22)
Category:
answer to question

  1. Indeed, overlap is better.

  2. The sentence below Eq 9 was indeed a remnant from the previous version of the paper, that was intended to be removed. Thanks for spotting this oversight.

  3. That Eq 11 is not only sufficient but also necessary can be shown by rewriting the projector using tensors that contain the missing columns to complement the MPS tensors in left canical form to a full unitary matrix, ie. tensors that contain a basis for the null space of A_L^dagger, often called V_L in older literature on this topic. We added a footnote to explain this.

  4. the tile symbol ‘~’ is meant to represent ‘is proportional to’. This is used commonly enough for us to be comfortable keeping it as it is.

  5. The difference in fidelity is remarked upon already. It is only significant if precision is of the utmost importance. However we don’t systematically argue this by looking at the differences in observable quantities (expectation values, correlation lengths,…) between local and variational truncation. Instead relying completely on the slightly better fidelity.

  6. It was indeed implied in the text that applying the sublattice rotation on one of the two ferromagnetic fixed points (which can be obtained via vumps), makes it equivalent to one of the antiferromagnetic fixed points (which cannot be obtained via vumps due to the alternating structure when applying the transfer matrix). We have added a sentence to make this statement explicit.

  7. Indeed, as you rightly said, the computational complexity is better for the variational method, regardless of substructure. However, in our experience, if one has an MPS that needs to be truncated without substructure, it is likely already in canonical form or one has already performed other operations of compational cost O(chi_2^3 d), and local truncation is therefore not more expensive. The problem we encountered, for which this method was devised, was the problem of having to truncate an MPS with some substructure, for which creating the full MPS tensor would yield an object that is simply too big to even fit in memory, thus prohibiting local truncation. We therefore focused on this particular situation, believing that this would be the most relevant setting also for other practitioners in the field.

Anonymous on 2021-02-12  [id 1228]

(in reply to Bram Vanhecke on 2021-02-11 [id 1227])

Thank you to the authors for the additional explanations, and for taking into account the points I made.

---

## Round 2 · Referee Report · Anonymous (Referee 3) · 2021-1-31

Report

The authors sufficiently addressed my raised points and I recommend the manuscript for publication

---

## Editorial Decision

published